health and disease and epidemiology

pooling, probabilistic model, COVID-19, qRT-PCR

**Author for correspondence:**
Uri Obolski
e-mail: uriobols@tauex.tau.ac.il

# An accurate model for SARS-CoV-2 pooled RT-PCR test errors

Yair Daon[1,2], Amit Huppert[1,3,†] and Uri Obolski[1,2,†]

[1]School of Public Health, The Faculty of Medicine, and [2]Porter School of the Environment and Earth Sciences, The Faculty of Exact Sciences, Tel Aviv University, Tel Aviv, Israel
[3]The Biostatistics and Biomathematics Unit, The Gertner Institute for Epidemiology and Health Policy Research, Sheba Medical Center, Tel Hashomer, 52621 Ramat Gan, Israel

UO, 0000-0001-7594-9745

Pooling is a method of simultaneously testing multiple samples for the presence of pathogens. Pooling of SARS-CoV-2 tests is increasing in popularity, due to its high testing throughput. A popular pooling scheme is Dorfman pooling: test $N$ individuals simultaneously, if the test is positive, each individual is then tested separately; otherwise, all are declared negative. Most analyses of the error rates of pooling schemes assume that including more than a single infected sample in a pooled test does not increase the probability of a positive outcome. We challenge this assumption with experimental data and suggest a novel and parsimonious probabilistic model for the outcomes of pooled tests. As an application, we analyse the false-negative rate (i.e. the probability of a negative result for an infected individual) of Dorfman pooling. We show that the false-negative rates under Dorfman pooling increase when the prevalence of infection decreases. However, low infection prevalence is exactly the condition when Dorfman pooling achieves highest throughput efficiency. We therefore urge the cautious use of pooling and development of pooling schemes that consider correctly accounting for tests' error rates.

## 1. Introduction

Reverse transcription polymerase chain reaction (RT-PCR) testing is the backbone and gold standard of infection surveillance across the globe, and a key component in the fight against the COVID-19 pandemic. One major application of this technology is the rapid testing of contacts of positive individuals in order to cut transmission chains. Moreover, in some countries, RT-PCR testing has become a routine procedure of surveillance, testing incoming air passengers in airports [1,2], school students [3], or even the entire adult population [4].

†Equal contribution.

As such, the need for large-scale testing has resulted in the application of techniques to increase the throughput of RT-PCR tests [5–10]. These techniques, often termed *pooling*, or group testing, have originated in the 1940s in the seminal work of Dorfman [5,11]. Under Dorfman pooling, one selects $N$ individuals and performs a single RT-PCR test on their combined (*pooled*) samples. If the pooled test yields a positive result, then each individual is retested separately; otherwise, everyone is declared negative. The throughput efficiency of Dorfman pooling has been demonstrated empirically [5] and its error rates thoroughly investigated [12–14].

The study and practice of pooling has advanced considerably since Dorfman's original paper, with several other pooling techniques being applied for SARS-CoV-2. For example, in *recursive* pooling [13,15], if the first pooled test is positive, the pool is split into two and the splitting process repeats. Otherwise, all pool members are declared negative. In *Matrix* pooling [8] a population of size $N = mn$ is arranged in a matrix of dimensions $m \times n$. Each row and column are then pooled and individuals in the intersection of positive rows and columns are tested separately. Other, more sophisticated pooling schemes exist [16], but studies investigating the underlying assumptions of pooling are surprisingly lacking.

Studies focused on pooling for SARS-CoV-2 are in a consensus that sample dilution effects [17,18] are not a concern, even for pools as large as 64 individuals [5,6,9,19,20]. Consequently, studies assumed that the probability of a true-positive (the test's *sensitivity*) does not depend on the number of infected samples in the pool, but rather on the existence of at least one such sample. Thus, the probability of a positive result in a pooled test has been assumed identical for a pool with one sample from an infected individual and, e.g. five such samples. This assumption is common in the group testing literature [6,12,13], as well as in more specific, SARS-CoV-2 focused studies [14,21]. In this study we challenge this commonly made assumption and demonstrate how using a more accurate probabilistic model affects the estimation of false-negative rates for Dorfman pooling.

Our analysis has several implications: previously estimated optimal Dorfman pool size [12] should be revisited in general and specifically for the case of SARS-CoV-2, as it is influenced by the prevalence of infection. The error rates of other pooling schemes in use should also be revisited [10,13]. Moreover, studies that seek precise error models should be conducted for other pathogens as well, since the standard error model could prove flawed. Specifically, all assumptions should be scrutinized, including (e.g.) the independence of tests, an extremely widespread assumption in the group testing literature [12,13] which dates back to Dorfman's original paper [11]. Finally, new methods should be developed in order to take advantage of more realistic models.

## 2. Methods

Formally, we consider a pool containing $N$ individuals $\{1, \ldots, N\}$. We denote the true infection state $\theta \in \{0,1\}^N$, so individual $i$ is infected if $\theta_i = 1$. The RT-PCR test's sensitivity (true-positive rate) is denoted $S_e$, and the test's specificity (true-negative rate) is denoted $S_P$. Pooled test result (data) is denoted $\mathbf{d} \in \{0, 1\}$, where $\mathbf{d} = 0$ if the test returned a negative result.

### 2.1. The common assumption

Previous studies of pooling schemes assumed that the false-negative probability does not depend on the number of infected samples, but merely on the existence of at least one such sample in a pool [12,13]. Current studies of pooling in the context of SARS-CoV-2 also employ a similar assumption [14,21]. Explicitly, these studies assume

$$\mathbb{P}(\mathbf{d} = 0|\theta) = \begin{cases} 1 - S_e & \exists i \text{ such that } \theta_i = 1 \\ S_P & \text{otherwise.} \end{cases} \tag{2.1}$$

Below, we refer to equation (2.1) as the *common assumption*.

### 2.2. Refuting the common assumption

We refute the common assumption with experimental data collected from [22], and summarized in table 1. There, the authors investigate Dorfman pooling and, regardless of the pooled test result, follow up and test each pool member separately. We focus on 128 pools for which at least one subsequent separate test was positive—of which 29 pooled tests were negative and 99 positive. In the data cited in [22], of the 29 negative

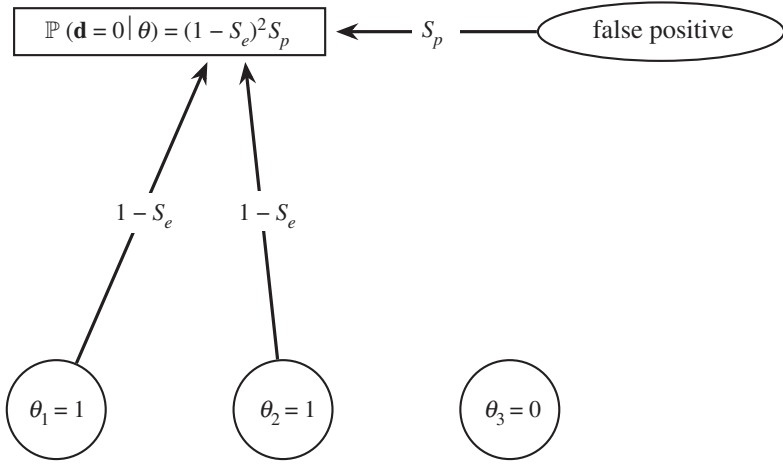

**Figure 1.** Illustration of the revised probabilistic model. A pool contains individuals {1, 2, 3} with state $\theta = (1, 1, 0)$ (i.e. individuals 1 and 2 are infected). A negative pooled test ($\mathbf{d} = 0$) occurs when three detection paths fail: a false-negative occurs for individuals 1 and 2, each with probability $(1 - S_e)$. Additionally, no false-positive detection occurs, with probability $S_p$. Individual 3 is not infected and does not contribute to the probability of the pooled test result.

**Table 1.** Contingency table of data from [22].

|                                   | negative pool | positive pool |
|-----------------------------------|:-------------:|:-------------:|
| no. subsequent positives = 1      | 24            | 42            |
| no. subsequent positives $\geq$ 1 | 5             | 57            |
| total                             | 29            | 99            |

pools, subsequent separate testing yielded a single positive result in 24. By contrast, of the 99 positive pools, 42 yielded a single positive test upon subsequent separate testing.

The data in table 1 allows us to test the following null hypothesis $H_0$: The probability of a pooled false-negative is equal for pools with one subsequent positively tested member and pools with two or more such members. $H_0$ is a direct consequence of the common assumption, and rejecting $H_0$ implies the common assumption is not realistic, at least for SARS-CoV-2.

We apply Fisher's exact test for the presence of more than one positive individual in correctly identified pools. Fisher's test yields an increased odds ratio of 6.4, 95% CI (2.2, 23.4), with a $p$-value $\approx 10^{-4}$. Thus, we reject $H_0$, refuting the common assumption.

## 2.3. Our model

Since the essence of the refuted common assumption is that amplification of all samples occurs only once, we assume a more realistic model: amplification of viral RNA succeeds or fails for each sample independently. Furthermore, according to [12–14,21], a false-positive does not depend on the number of negative samples in a pool either. For lack of data pointing otherwise, we incorporate this assumption into our model with a small modification. We do assume that a false amplification can occur only once per pool. However, we also assume false amplification is independent of any other correct amplification. Specifically, it is possible that every correct amplification fails *and* an erroneous one occurs simultaneously. This assumption is somewhat specific for the current application of screening for SARS-CoV-2 via RT-PCR. For example, cross-reactivity with other coronaviruses would have violated this assumption. However, cross-reactivity was ruled out in [23]. These assumptions lead to the following model, illustrated in figure 1 and summarized in equation (2.2).

$$\mathbb{P}(\mathbf{d} = 0 | \theta) = S_P \prod_{i=1}^{N} (1 - S_e)^{\theta_i} = S_P (1 - S_e)^{\sum_i \theta_i}. \tag{2.2}$$

## 2.4. Application: scheme false-negative rate

We calculate the false-negative rate for a single *infected* individual, henceforth referred to as 'Donald', under our model and under the common assumption. We distinguish three types of false-negative events when performing pooling. A *single test*'s false-negative is the event of a negative result upon testing Donald separately, i.e. in an RT-PCR test without pooling. A *pooled* false-negative occurs when a pooled test containing Donald's sample (and other samples) yields a negative result, i.e. the pooling fails to detect at least one positive result. Lastly, a *scheme* false-negative occurs when an entire pooling scheme fails to identify Donald as infected. Our first task is to calculate Dorfman's scheme false-negative rate. Equivalently, we ask: what is the probability of not identifying Donald as infected under Dorfman pooling?

Denote the prevalence of infection in the (tested) population $q$. We denote Donald as individual 1, so that $\theta_1 = 1$. Then

$$
\begin{aligned}
\mathbb{P}(\mathbf{d} = 0 | \theta_1 = 1) &= \sum_{\theta_2,\dots,\theta_N} \mathbb{P}(\theta_2, \dots, \theta_N) \mathbb{P}(\mathbf{d} = 0 | \theta_1 = 1, \theta_2, \dots, \theta_N) \\
&= \sum_{k=0}^{N-1} \mathbb{P}\left( \sum_{i=2}^{N} \theta_i = k \right) \mathbb{P}\left( \mathbf{d} = 0 | \theta_1 = 1, \sum_{i=2}^{N} \theta_i = k \right) \\
&= \sum_{k=0}^{N-1} \binom{N-1}{k} q^k (1-q)^{N-1-k} S_p (1 - S_e)^{1+k} \\
&= S_p (1 - S_e) \sum_{k=0}^{N-1} \binom{N-1}{k} (q(1-S_e))^k (1-q)^{N-1-k} \\
&= S_p (1 - S_e)(1 - q S_e)^{N-1}.
\end{aligned}
\tag{2.3}
$$

If the pooled test yields a positive result, Donald is tested separately. Taking a conservative stand, it is assumed that such a simple procedure poses no risk of introducing contaminant RNA. Therefore, the separate test yields a positive result with probability $S_e$.

We calculate the probability that Donald is mistakenly identified as not infected, henceforth referred to as the scheme's false-negative rate and denoted $S_{fn}$. In order to correctly identify an infected individual as infected, both pooled and separate tests have to yield a positive result. Thus, the scheme's false-negative rate is

$$
\begin{aligned}
S_{fn} &:= 1 - S_e \mathbb{P}(\mathbf{d} = 1 | \theta_1 = 1) \\
&= 1 - S_e [1 - S_p (1 - S_e)(1 - q S_e)]^{N-1}.
\end{aligned}
\tag{2.4}
$$

### 2.4.1. Comparison metric

The single test false-negative rate $1 - S_e$ and scheme false-negative rate $S_{fn}$ are compared via

$$
E_{exact} := \frac{S_{fn} - (1 - S_e)}{1 - S_e} \times 100\%.
\tag{2.5}
$$

$E_{exact}$ is the percentage increase in the pooling scheme false-negative rate, relative to the single test false-negative rate.

According to the common assumption, the scheme false-negative rate is $1 - S_e^2$. A straight forward calculation shows that this implies the percentage increase in scheme false-negative rate is $E_{common} := 100\% \cdot S_e$.

## 3. Results

### 3.1. Scheme false-negative

We plot $E_{exact}$ for varying prevalence $q$ and sensitivity $S_e$ values, and make the comparison with $E_{common}$. As recommended by [5], we apply different pool sizes $N$, for different prevalence values. We consider a specificity of $S_p = 0.95$ [5] and a range of reasonable sensitivity and prevalence values [23–26]. We observe that the increase in the scheme false-negative rate (compared to $1 - S_e$), expressed by $E_{exact}$, is at least 60% (figure 2).

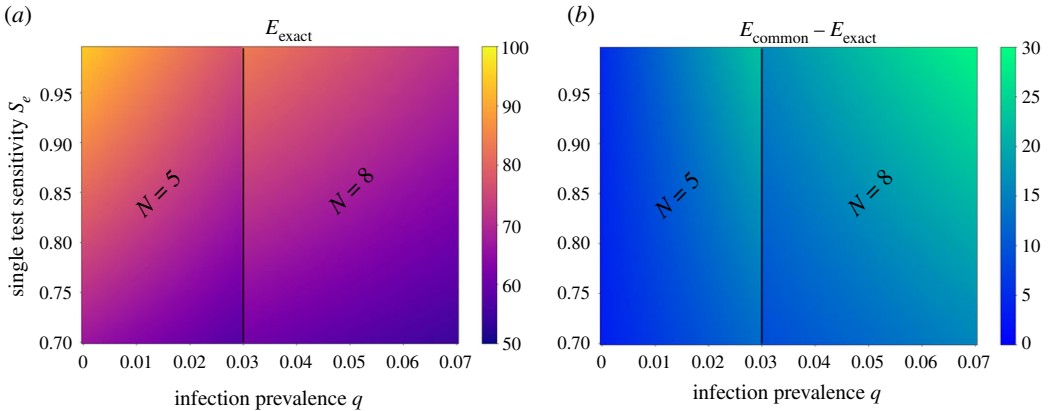

**Figure 2.** Relative increase in Dorfman pooling false-negative rates $E_{exact}$. (a) Colours represent $E_{exact}$, the relative percentage increase in the scheme false-negative rates relative to the single test false-negative rates equation (2.5). (b) Colours represent the difference between $E_{common}$ and $E_{exact}$. The infection prevalence $q$ is varied on the x-axis, while the test sensitivity is varied on the y-axis. Pool size $N$ was chosen according to $q$ as in [5]. Panel a shows that $E_{exact}$ is largest for low prevalence values $q$. The difference between $E_{common}$ and $E_{exact}$ can be as large as 30%, as seen in b.

Interestingly, an increase in infection prevalence monotonically decreases the scheme false-negative rate, as can also be easily seen from equation (2.4). For the chosen parameter ranges, the increase in the single test false-negative rates increases the relative error $E_{exact}$. These effects can be seen in figure 2a, upon conditioning on pool size. Extending the range for $S_p$ yields no qualitative differences. We further compare $E_{exact}$ to $E_{common}$, showing the discrepancy changes as a function of both prevalence and the single test sensitivity (figure 2b).

## 4. Discussion

In this study, a novel probabilistic model is developed to capture realistic features of the outcomes of pooled RT-PCR tests, parameterized for SARS-CoV-2. Contrary to the common assumption, we assume, based on data (table 1), that multiple infected individuals increase the likelihood of a positive pooled test result. A direct consequence of our model is that low values of infection prevalence increase the false-negative rates of Dorfman pooling. Importantly, our results remain qualitatively similar under varying parameter values, in the observed ranges [23–25,27] (figure 2). Therefore, our results give rise to a conflict: low infection prevalence leads to high efficiency of Dorfman pooling [5] but also increases false-negative rates.

As the COVID-19 pandemic progresses, the infection prevalence in various tested populations undergoes frequent changes. Hence, as our results suggest, pooling schemes employed for mass testing should be used with caution in populations where infection rates are low. Such mass-tested populations often include air travel passengers [28], nursing homes [5], or presymptomatic and asymptomatic individuals [29], and can be crucial for controlling outbreaks [30].

Furthermore, an especially important consideration in designing pooling schemes is explicitly taking into account the intrinsic RT-PCR error rates. As illustrated above, analysing the expected error rates of pooling schemes under a realistic error model is readily achievable. Designing pooling schemes that optimize testing while accounting for error rates is, however, considerably more involved. Nonetheless, various algorithms explicitly incorporate error rates into their pooling optimization processes [12,13], including a recent study by our group which employed the herein described error model [31]. In our proposed algorithm, we have used an experimental design criterion to create an algorithm which iteratively performs pooling, updating its information based on previous results. In doing so, our pooling scheme shows potential in terms of both reducing false-negative and false-positive rates, as well as increasing efficiency, although it has not yet been implemented in real-world settings.

To conclude, pooling is an important technique that can increase testing throughput in a cost-effective manner. Nevertheless, care must be given to pooling schemes' false-negative rates, especially under low infection prevalence settings.

Authors' contributions. Data and relevant code for this research work are stored in GitHub: https://github.com/yairdaon/errorates and have been archived within the Zenodo repository: https://doi.org/10.5281/zenodo.5510563 [32]. Y.D., A.H. and U.O. designed the study. Y.D. wrote code.

Competing interests. We declare we have no competing interests.

Funding. Y.D. was supported by a post-doctoral fellowship from the Tel Aviv University Center for Combating Pandemics and the Raymond and Beverly Sackler dean's post-doctoral fellowship.

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
