## [Peer Review File · Royal Society Open Science]

Review History

RSOS-210704.R0 (Original submission)

Review form: Reviewer 1

Is the manuscript scientifically sound in its present form?

Yes

Are the interpretations and conclusions justified by the results?

Yes

Is the language acceptable?

Yes

Do you have any ethical concerns with this paper?

No

Have you any concerns about statistical analyses in this paper?

No

Recommendation?

Accept with minor revision (please list in comments)

Comments to the Author(s)

The authors challenge the pooling method used today in testing for SARS-CoV-19 and propose a more robust model. As COVID-19 is here to stay for a while longer this method can improve the accuracy of detection and prevention of spread. However, some minor changes need to be made before this paper can be published.

The introduction is extremely slim it is hard to understand from it why we need a new model, as all the authors say is how widely used and very accurate is the current pooling model. I would advise the authors to add a sentence or two at least on what is the main problem with the pooling method and why we need a different model instead of just saying that they are challenging the current method.

The first paragraph of the discussion is a bit unclear with sentences detached from one another. The sentence starting with "Specifically" is detached from the rest of the paragraph and it is unclear why it is there. I would suggest rewriting the whole paragraph to make it clear and concise.

The authors published all their code on github as requested by the journal. However, they leave the reader to fend for himself and figure out what is what. This makes the code less accessible and decreases the chances of the model being implemented or replicated. The authors should add a README.md that explains a bit about the code in the repository and what it's meant for.

Specific comments:

Page 3 Line 9: There are two "the" one should be erased.

Page 7 Line8: where it says "can also be see from (4)" it is unclear what 4 refers to - a reference or an equation. Please correct it.

Decision letter (RSOS-210704.R0)

Dear Dr Obolski

The Editors assigned to your paper RSOS-210704 "An Accurate Model for SARS-CoV-2 Pooled RT-PCR Test Errors" have now received comments from reviewers and would like you to revise the paper in accordance with the reviewer comments and any comments from the Editors. Please note this decision does not guarantee eventual acceptance.

Please submit your revised manuscript and required files (see below) no later than 21 days from today's (ie 08-Sep-2021) date. Note: the ScholarOne system will 'lock' if submission of the revision is attempted 21 or more days after the deadline. If you do not think you will be able to meet this deadline please contact the editorial office immediately.

on behalf of Professor Joshua Ross (Associate Editor) and Malcolm White (Subject Editor)
openscience@royalsociety.org

Associate Editor Comments to Author (Professor Joshua Ross):

Associate Editor: 1

Comments to the Author:

In addition to the comments of the Reviewer, can you please address the following before we reconsider your paper for publication:

1. Typo on Line 15 of Page 3;
2. Formatting of quotation marks on Line 28 of Page 5;
3. Formatting of parentheses in Equation (3) on page 6;
4. Add % to Equation (5) and line 40 of the same page;
5. Ensure Code base is well documented;
6. Clear up what the increase is with respect to on Line 6 of Page 7;
7. It feels strange that you have simply highlighted a potential issue, and trade offs, without specifying any guidance as how best to perform pool testing given your findings. I feel such discussion and interpretation is necessary. You do cite your own paper which appears to have used optimal experimental design to determine an optimal design for such -- should you compare that to these results here?

Reviewer comments to Author:

Reviewer: 1

Comments to the Author(s)

The authors challenge the pooling method used today in testing for SARS-CoV-19 and propose a more robust model. As COVID-19 is here to stay for a while longer this method can improve the

accuracy of detection and prevention of spread. However, some minor changes need to be made before this paper can be published.

The introduction is extremely slim it is hard to understand from it why we need a new model, as all the authors say is how widely used and very accurate is the current pooling model. I would advise the authors to add a sentence or two at least on what is the main problem with the pooling method and why we need a different model instead of just saying that they are challenging the current method.

The first paragraph of the discussion is a bit unclear with sentences detached from one another. The sentence starting with "Specifically" is detached from the rest of the paragraph and it is unclear why it is there. I would suggest rewriting the whole paragraph to make it clear and concise.

The authors published all their code on github as requested by the journal. However, they leave the reader to fend for himself and figure out what is what. This makes the code less accessible and decreases the chances of the model being implemented or replicated. The authors should add a README.md that explains a bit about the code in the repository and what it's meant for.

Specific comments:

Page 3 Line 9: There are two "the" one should be erased.

Page 7 Line 8: where it says "can also be see from (4)" it is unclear what 4 refers to - a reference or an equation. Please correct it.

===PREPARING YOUR MANUSCRIPT===

===PREPARING YOUR REVISION IN SCHOLARONE===

Author's Response to Decision Letter for (RSOS-210704.R0)

See Appendix A.

Decision letter (RSOS-210704.R1)

Dear Dr Obolski,

It is a pleasure to accept your manuscript entitled "An Accurate Model for SARS-CoV-2 Pooled RT-PCR Test Errors" in its current form for publication in Royal Society Open Science. The comments of the reviewer(s) who reviewed your manuscript are included at the foot of this letter.

COVID-19 rapid publication process:

We are taking steps to expedite the publication of research relevant to the pandemic. If you wish, you can opt to have your paper published as soon as it is ready, rather than waiting for it to be published the scheduled Wednesday.

This means your paper will not be included in the weekly media round-up which the Society sends to journalists ahead of publication. However, it will still appear in the COVID-19 Publishing Collection which journalists will be directed to each week (<https://royalsocietypublishing.org/topic/special-collections/novel-coronavirus-outbreak>).

If you wish to have your paper considered for immediate publication, or to discuss further, please notify openscience_proofs@royalsociety.org and press@royalsociety.org when you respond to this email.

on behalf of Professor Joshua Ross (Associate Editor) and Malcolm White (Subject Editor)
openscience@royalsociety.org

Associate Editor Comments to Author (Professor Joshua Ross):

Associate Editor

Comments to the Author:

All comments have been adequately addressed. The issue with the quotation marks is the direction of the opening quotation marks.

Appendix A

Dear Professor Joshua Ross,

Thank you very much for considering a revision of our manuscript titled “An Accurate Model for SARS-CoV-2 Pooled RT-PCR Test Errors” (**Manuscript ID RSOS-210704**) for *Royal Society Open Science*.

Please find the reviewers’ and your comments (**in bold**) with a point-by-point response (regular font). We have answered all the points raised and hope that you now find the manuscript suitable for publication.

Yours sincerely, on behalf of all authors,

Uri Obolski

Associate Editor

1. Typo on Line 15 of Page 3;

Fixed

2. Formatting of quotation marks on Line 28 of Page 5;

We apologize, but the quotation marks seem normal when we observe the PDF. We have tried using a different font for them and hope it looks better now.

3. Formatting of parentheses in Equation (3) on page 6;

Fixed

4. Add % to Equation (5) and line 40 of the same page;

Done

5. Ensure Code base is well documented;

Added a README.md file and comments in the code (see below, in our answer to reviewer 1).

6. Clear up what the increase is with respect to on Line 6 of Page 7;

Fixed

7. It feels strange that you have simply highlighted a potential issue, and trade offs, without specifying any guidance as how best to perform pool testing given your findings. I feel such discussion and interpretation is necessary. You do cite your own paper which appears to have used optimal experimental design to determine an optimal design for such -- should you compare that to these results here?

We agree, we now highlight further issues that may arise when employing the common model in the introduction. We also added a paragraph to the end of the discussion, where we shortly discuss our algorithm, as well as the topics raised by the reviewer.

Reviewer comments to Author:

Reviewer: 1

Comments to the Author(s)

The authors challenge the pooling method used today in testing for SARS-CoV-19 and propose a more robust model. As COVID-19 is here to stay for a while longer this method can improve the accuracy of detection and prevention of spread. However, some minor changes need to be made before this paper can be published.

The introduction is extremely slim. It is hard to understand why we need a new model, as all the authors say is how widely used and very accurate is the current pooling model. I would advise the authors to add a sentence or two at least on what is the main problem with the pooling method and why we need a different model instead of just saying that they are challenging the current method.

We agree that the Introduction was not extensive enough and therefore have substantially revised and improved it. We also now emphasize the problems with current models both in the Introduction and in the Discussion.

The first paragraph of the discussion is a bit unclear with sentences detached from one another. The sentence starting with “Specifically” is detached from the rest of the paragraph and it is unclear why it is there. I would suggest rewriting the whole paragraph to make it clear and concise.

We have now revised the Discussion to enhance readability and hope that the new version is clearer.

The authors published all their code on github as requested by the journal. However, they leave the reader to fend for himself and figure out what is what. This makes the code less accessible and decreases the chances of the model being implemented or replicated. The authors should add a README.md that explains a bit about the code in the repository and what it’s meant for.

We apologize that the github repository was not sufficiently organized and accessible. We have now added comments in the code as well as a README.md file to enhance reproducibility.

Specific comments:

Page 3 Line 9: There are two “the” one should be erased.

Fixed

Page 7 Line 8: where it says “can also be see from (4)” it is unclear what 4 refers to – a reference or an equation. Please correct it.

Added “equation” to all such occurrences.